# Osteopontin: The Molecular Bridge between Fat and Cardiac–Renal Disorders

**DOI:** 10.3390/ijms21155568

**Published:** 2020-08-04

**Authors:** Elena Vianello, Marta Kalousová, Elena Dozio, Lorenza Tacchini, Tomáš Zima, Massimiliano Marco Corsi Romanelli

**Affiliations:** 1Department of Biomedical Sciences for Health, Università degli Studi di Milano, 20133 Milan, Italy; elena.dozio@unim.it (E.D.); lorenza.tacchini@unimi.it (L.T.); mmcorsi@unimi.it (M.M.C.R.); 2Institute of Medical Biochemistry and Laboratory Diagnostic, First Faculty of Medicine, Charles University and General University Hospital in Prague, 12108 Prague, Czech Republic; marta.kalousova@lf1.cuni.cz (M.K.); zimatom@cesnet.cz (T.Z.); 3U.O.C. SMEL-1 of Clinical Pathology, IRCCS Policlinico San Donato, 20097 San Donato Milanese, Italy

**Keywords:** osteopontin (OPN), visceral adipose tissue (VAT), obesity, cardiovascular diseases (CVDs), renal disorders, chronic kidney disease (CKD)

## Abstract

Osteopontin (OPN) is a multifaceted matricellular protein, with well-recognized roles in both the physiological and pathological processes in the body. OPN is expressed in the main organs and cell types, in which it induces different biological actions. During physiological conditioning, OPN acts as both an intracellular protein and soluble excreted cytokine, regulating tissue remodeling and immune-infiltrate in adipose tissue the heart and the kidney. In contrast, the increased expression of OPN has been correlated with the severity of the cardiovascular and renal outcomes associated with obesity. Indeed, OPN expression is at the “cross roads” of visceral fat extension, cardiovascular diseases (CVDs) and renal disorders, in which OPN orchestrates the molecular interactions, leading to chronic low-grade inflammation. The common factor associated with OPN overexpression in adipose, cardiac and renal tissues seems attributable to the concomitant increase in visceral fat size and the increase in infiltrated OPN^+^ macrophages. This review underlines the current knowledge on the molecular interactions between obesity and the cardiac–renal disorders ruled by OPN.

## 1. Introduction

Osteopontin (OPN) is an extracellular matrix protein first identified in bone tissue and has pleiotropic functions due to its ubiquitous expression in the main organs and apparatuses [1]. Indeed, OPN has been physiologically found in bone cells as well as in adipocytes, cardiomyocytes, renal and vascular cells [2,3]. In each anatomical compartment, OPN mediates its primary functions involved in the regulation of T-cells, macrophage recruitment and infiltration, as well as the inhibition of local calcification signaling [4,5]. OPN exists in two main isoforms, namely, an intracellular and a soluble form, which mediate physiological and pathological signaling [6,7].

Acting as a matricellular immunomodulatory cytokine, OPN intervenes in different cell activities, including cell proliferation and inflammatory responses, through the binding of its corresponding receptors [8]. The matricellular proteins are a class of extracellular matrix (ECM) proteins that achieve their functions by binding to ECM proteins, like integrins, or to the cell surface receptors or other molecules, including cytokines and proteases, which, in turn, interact with the cell surface by influencing cell adhesion, migration, proliferation, differentiation and apoptosis [9]. Normally, few matricellular proteins are expressed in adult tissues, but their expression increases during development and under injury or pathology [10]. In this regard, as part of the integrin and CD44 ligands, OPN is one of the main mediators of ECM–cell interactions [11]. The binding between the matricellular OPN and ECM components, including collagen, laminin, fibronectin, etc., sends intracellular signals to the targeted tissues [12,13]. Because the integrins do not have enzymatic functions, to transmit their signal they must trigger the downstream molecules, such as the immune components [1]. Noteworthy is that the hydrodynamic stress sear affecting the renal, cardiac and fat tissues in case of obesity can promote detrimental mechanical forces, inducing OPN expression and its release in these tissues [14]. Due to the multifunctional properties of OPN and its ubiquitous expression in the body, its pivotal role in coexistent disorders like obesity, renal dysfunctions and cardiovascular diseases (CVDs) is currently debated [15,16], in which simultaneous etiologic factors can promote OPN signaling deregulation and aggravate the cross-disorders, like cardiac or renal–fat diseases [15,16,17]. Indeed, it is notable that the main common factor highlighted as a promoter of the progression and development of cardiovascular and renal disorders is the increase in visceral fat mass [18,19,20]. Visceral adipose tissue (VAT), which extends during overweight and obesity, is associated not only with the development of type 2 diabetes or hypertension, but also it is recognized as a well-known risk factor for heart failure (HF), coronary artery diseases, valve diseases and for all main events associated with the loss of renal functions [21,22]. In this field, researchers showed the pivotal role of OPN as a pro-inflammatory cytokine able to regulate the main intracellular signaling associated with the decline in fat [22,23], heart [24] and kidney functions [25,26]. In the present review, we first outline the current knowledge regarding OPN molecular signaling in the physiological and pathological conditions associated with obesity. We further focus on the role of OPN as a possible molecular cross-biomarker for obesity-associated disorders.

## 2. Osteopontin (OPN) Structure and Physiological Functions

Osteopontin, also known as secreted phosphoprotein 1 (SPP-1), uroprotein or early T lymphocyte activation-1 (Eta-1), is a glycophosphoprotein that is highly phosphorylated, composed of 314 amino acids [1,27]. The molecular weight of OPN ranges between 41 and 75 kDa, due to the multiple post-translational changes, including O-linked glycosylation, phosphorylation, sialylation and tyrosine sulfation, which could modify the OPN responses in the tissue [6,27]. These modifications are converted to three distinct human OPN transcripts: OPN-a (the full-length), OPN-b (missing of exon 5) and OPN-c (missing of exon 4), although little it is known about their role in the organs [7]. OPN was discovered in 1985 as a major sialoprotein in bone, a member of the small integrin-binding ligand N-linked glycoprotein (SIBLING) family of proteins, implicated in bone mineralization and remodeling [28]. In addition to bone metabolism, OPN acts as a regulator of the immune response. Indeed, this protein has chemotactic properties for macrophages, dendritic and T cells, thanks to its interactions with multiple surface proteins localized in this cell’s targets [27]. The main interactions occur through integrin receptor binding that directly activates the nuclear factor kappa B (NFkB) in the intracellular space [29]. Moreover, OPN also can regulate the immune system through surface interaction with the CD44 receptor, able to stimulate T-cell chemotaxis, adhesion, and inhibition of interleukin(IL)-10 release by the macrophages [29]. These interactions are principally mediated by the two different terminal zones (the N and C terminal, respectively), aimed to bind different ligands [30]. The C terminal interacts with two heparin molecules or CD44 variants, whereas the N terminal is involved in the interaction with integrin receptors [1]. These functional domains are conserved in human and other species, implying important shared activities [1]. The C-terminal fragment is involved in macrophage chemotaxis [1,5]. The N terminal is especially implicated in the regulation of hematopoietic progenitor cell homing and in the secretion of interferon (IFN)-γ by the T cells [31]. Due to the multiplicity of the OPN ligands, including heparin, surface receptors, intracellular signaling particles and calcium, this protein is reputed as one of the major bio-controllers of the body system. In addition to the well-known control in bone regulation and immune cells, the fundamental actions of OPN in adipose tissue, heart and kidney metabolism is well reported [32,33]. Indeed, OPN physiologically regulates the developmental processes, local tissue remodeling and healing, as well as the senescence in adipocytes, cardiac and renal cells [26]. On the contrary, the aberrant expression of OPN is directly associated with obesity-related disorders, such as type 2 diabetes, cardiac and renal dysfunctions [20,34]. Moreover, it is reported that the upregulation of OPN expression in immune systems leads to worsening of chronic inflammatory conditions, including atherosclerosis and vascular dysfunction [34]. OPN regulation is at the “cross roads” of adipose tissue depots as well as heart and renal cells. Indeed, different studies suggest the multiple roles of this matricellular protein in mediating cross-disorders [33,35]. Moreover, working as a matricellular component, OPN is part of a proteinaceous network that mechanically support and organize the extracellular space of multicellular organs, including adipose tissue, hearts and kidneys, mediating and integrating their molecular signals and cellular responses [17]. OPN serves as link between neighboring cells, acting as dynamic regulator of their microenvironmental activities [34].

Understanding the molecular cross-talk regulated by OPN could be a novel starting point for a new view of cross-disorders.

## 3. OPN as a Matricellular Protein in Fat, Cardiac and Renal Tissues

Several lights of evidence reported that adipose tissue is a dynamic organ with constant remodeling necessary to meet the demand of the ever-changing fat in normal and pathological conditions [36]. Indeed, adipocytes undergo different mechanical stresses due to free fatty acid uptake, which promote morphological changes inside the tissue [37]. This ability of the fast remodel is the result of the coordinated activities of resident adipocytes, immune cells, endothelial cells and fibroblasts [36]. In normal conditions, these groups of cells oppose against pro-inflammatory stimuli during tissue expansion, which become harmful in uncontrolled chronic fat expansion beyond the tissue’s capacity, leading to persistent fibrosis, hypoxia, cell death and senescence [36,37]. This occurs in all kinds of metabolic disorders, including overweight, obesity and type 2 diabetes [37]. In this contest, adipocyte necrosis is triggered by the excess of ECM deposition and OPN expression that function as chemoattractant mediators, activating the classic pro-inflammatory M1 macrophages’ way and extent of inflammation and metabolic dysfunction [37,38]. Moreover, OPN as part of the ECM components in adipose tissue dramatically increased its expression in a murine model of a high fat diet, in genetically obese rodents and in obese humans [37,38,39,40]. OPN is also considered a modifier of ECM in adipose tissue, since it played a specific role in the pathogenesis of ECM dysregulation due to its direct involvement in the metalloproteinase activities and collagen deposition or resorption [36]. Notably, it is reported that, as a matricellular protein, OPN can mediate not only the ECM fat turnover but also that of cardiac and renal tissues [1,41]. In the heart, the ECM components create an intricate three-dimensional network, giving structural support and integrating extracellular signaling and cell responses [42]. When the mechanical forces become dangerous under pathological conditions like coronary occlusion, valve disorders and atherogenesis, the extent of the fibrotic remodeling as an outcome of the extent of the ECM deposition and accumulation is a common factor in myocardial diseases [43]. In normal myocardium, matricellular proteins are nominally expressed, but under cardiac injury they become overexpressed [44,45]. Left ventricular pressure overload resulting from systemic hypertension or vessel stenosis promotes both interstitial and perivascular fibrosis [46]. OPN as a secreted matricellular protein acts as a link between the EMC components and cardiac cells to modulate cell behavior [47,48,49,50]. In the heart, OPN regulates inflammation—both secreted and ECM bound—promoting anti-apoptotic signals. In the context of chronic cardiac overload, OPN mediate the ECM responses via macrophage recruitment and fibroblast activation in the site of injury [51]. These kinds of ECM events mediated by OPN occur also in renal tissue. Indeed, ECM accumulation and increased stiffness due to mechanical forces in the kidneys lead to progressive replacement of the flexible parenchyma, reducing renal function and promoting chronic kidney disease (CKD) [52]. This is due to the connection between cardiac and renal disorders. Indeed, the decline in cardiac output promotes renal blood flow reduction and renal dysfunction causes cardiac overload, in which OPN plays a pivotal role [53]. As a matricellular protein, OPN is the first mediator associated with ECM degeneration, which stimulates the release of the local fibrogenic mediators and promotes the tissue’s failure. In renal inflammation and fibrosis, matricellular OPN is involved in macrophages and T-cell recruitment, contributing to propagate ECM deposition signaling [54,55]. Under chronic renal flow overload, angiotensin II upregulates transforming growth factor β (TGF-β) as well all the mediators associated with ECM turnover, including OPN [14]. In the light of this, it is plausible to consider OPN as a matricellular protein associated with the cross-disorders associated with cardiac and renal fat’s decline, as well as with macrophage activation and recruitment in the site of tissue damage.

## 4. OPN in Obesity

As previously reported, we know that adipose tissue metabolism and function are predominantly regulated by nonfat cells, including immune cells, endothelial cells and fibroblasts, as part of stromal cells. Among them, adipose tissue macrophages (ATMs) are considered the main controller of fat depots [56]. Indeed, ATMs in adipose tissue, particularly in visceral compartments, control the local adipocytes turnover during dynamic changes of the tissue, which take place physiologically in the course of lipid energy storage activities [16]. Interestingly, both in VAT and ectopic VAT depots, like adipose tissue present in intra-cardiac and renal organs, ATMs play a pivotal role in local inflammation, because in an abnormal extended fat mass, ATMs can organize themselves in crown-like structures (CLs) [57,58], to confine and delete damaged cells after mechanical injuries occurring in response to fat hypertrophy and/or hyperplasia [16,57]. Several lines of evidence indicate that the low-grade inflammation in obesity is caused by obesity-induced inflammation of the adipose tissue, which is primarily driven by ATMs [56]. Indeed, ATMs are the primary source of pro-inflammatory mediators inside the adipose tissue and directly implicated in the pathogenesis of obesity-associated disorders, including cardiac and renal diseases [59,60,61,62]. In the last years, the CLs have become synonymous with metabolic dysfunction, perhaps for the rate of adipocyte death in obesity causing inflammation [58,63,64]. Intriguingly, it is demonstrated that OPN is expressed predominantly by ATMs in adipose tissue and regulates the local inflammation and cell death [60]. At present, numerous studies have shown that OPN is the main adipokine involved in the recruitment and accumulation of ATMs in the tissue [22,56]. In rodent and human models of obesity, OPN is the principal cytokine overexpressed in adipose organs, especially by ATMs, and secondarily by other inflammatory cells, like dendritic, stromal and vascular cells, as well as adipocytes [56]. This is confirmed by the fact that in obese OPN-knockout rodents there is a significant reduction in high-fat diet-induced adipose tissue macrophage infiltration and inflammation [57]. For this reason, it is highlighted that VAT is the main source of OPN during obesity [65]. In addition, the metabolic changes orchestrated by OPN in VAT [66,67] advance the hypothesis of its role as a key regulator of the inflammatory processes of obesity-induced adipose tissue inflammation [64] and become a possible target for treatment of adipose tissue-associated disorders, such as cardiac and renal diseases [15,17].

## 5. Cardiac–Fat Diseases Ruled by OPN

A growing body of evidence suggests the “double-faced” role of OPN in mediating repair or tissue damage in the heart, due to its dual involvement in both acute and chronic responses [3]. As part of the inflammatory cytokines, OPN is necessary for the acute response to the insult and progressive healing that activates cardiac repair [3,68]. On the contrary, when the acute event becomes persistent, OPN can drive and propagate chronic inflammatory signaling [69]. Evidence from the literature suggests that OPN in acute events is protective, attenuating scar deposition after myocardial damage and promoting neovascularization, especially thanks to the shift in infiltrated macrophage polarization toward M2, aimed to resolve the local insult and reconstruct the connective tissue [3,70]. In contrast, when the injury does not resolve, the OPN level is clinically related with the majority of cardiovascular outcomes associated with HF and obesity [22]. In the healthy myocardium, the OPN protein is not expressed, but under mechanical stress, occurring during pressure overload, cardiac injury, hypoxia and overweight, as well as perivascular and interstitial cells increase the production and release of OPN, promoting fibrosis and failure [71,72]. In the unstressed heart, the molecular expression of OPN in ventricular cardiomyocytes is basal, but during cardiac hypertrophy promoted by hydrodynamic shear, both the OPN mRNA and protein levels increase, suggesting its involvement in matricellular signaling aimed to promote myocyte growth to contrast detrimental cell death [34,72]. Interestingly, in fibrotic lesions and healing wounds, it seems that myofibroblast differentiation is mediated by OPN. Moreover, it is reported that the OPN level increases in the plasma of patients affected by dilated cardiomyopathy, although the OPN plasma increase seems more related to patients with HF severity than to the type of cardiomyopathy [34]. Making matters worse is the concomitant presence of CVDs and abnormal body fatness. Pre-published research has highlighted that the pathogenesis of obesity-associated disorders like CVDs is driven by a disequilibrium of adipokines production in extended VAT [22]. During obesity, VAT accumulation increases in different anatomical depots, including epicardial and pericardial fats [73]. Under physiological conditions, all VAT depots act as helpful single organs, with safe metabolic activities aimed at accumulating a lipid surplus and releasing anti-inflammatory adipokines, such as adiponectin, to preserve body homeostasis [74]. On the contrary, in overweight and obese patients, VAT becomes dysfunctional, secreting different kinds of mediators that are both involved in fat and heart decline [74]. This is due to the intimate connection between the myocardium and a peculiar VAT depot, known as epicardial fat [74,75,76], as there is no fascia muscle separating their anatomical layer walls [75]. In dysfunctional epicardial fat, the release of pro-inflammatory adipokines, including tumor necrosis factor (TNF)-α, interleukin (IL)-6, leptin and monocyte chemoattractant protein-1 (MCP-1) [77], contributes to worsening the atherogenic processes affecting the coronary arteries and myocardium [77]. Moreover, there is a growing interest in the characterization of the immune fat lineage in obese patients. Different studies found a majority of CD68^+^ cells in adipose tissue immune-infiltrate and most of them are polarized toward the pro-inflammatory M1 state (CD68^+^/CD11c^+^) [78], overexpressing and secreting soluble osteopontin (sOPN), which reaches the myocardium and amplifies the cardiac pro-inflammatory responses [78]. Indeed, it is reported that CVD patients present higher OPN mRNA and protein levels, both in the heart and epicardial fat [79]. The concomitant role of OPN in a dysfunctional VAT and heart suggests their pivotal role in cardiac–fat disorders, although more studies are needed to elucidate OPN cross-cellular signals governing cardiac–fat inflammation.

In the view of the role of OPN mediating cardiac–fat disorders, little is known about the fibro-adipocyte progenitors (FAPs) in the heart and OPN expression. Several lines of evidence highlighted that a subclass of cardiac fibroblasts may possess stem potential [80]. Under physiological conditions, these kinds of cells are quiescent as multipotent cell reserves [81]; but, during cardiac injury, cardiomyocytes send stress signals that promote immune infiltration and the multipotent progenitors’ activation, increasing the cell proliferation rate, matrix deposition and cytokines secretion, including OPN [82]. Moreover, injured myofibroblasts go along the senescent pathway and secretes senescent-inducing proteins aimed at restoring cardiac function after heart damage [81]. Notably, the resident subset of cardiac FAPs with biopotential differentiation versus fibro or adipose lineages can intervene in cardiac aging through their differentiation into fibrotic or adipose cells [81]. Indeed, persistent injury in myofibroblasts leads to cardiac adipocyte expression, originating from cardiac FAPs [59]. In this view, a new study reported that adipose-derived OPN and FAPs in the heart can inhibit cardiac fibroblast senescence in a rodent model and promote a pro-fibrotic response. Intriguingly, OPN-null animals exhibited reduced fibrotic lesions and normal fibroblast senescence [83]. This could be explained by the fact that OPN^+^ macrophages play a fundamental role in the control of fibrogenic cell accumulation and activities, regulating cardiac fibrosis [83]. Indeed, under physiological conditions, satellite cells inhibit FAP differentiation into the adipo-lineage [84]. However, during chronic disorders affecting cardiac cells, FAPs are the main contributors of ectopic VAT deposition and scar formation [84]. Notably, it is reported that the decrease in FAP accumulation correlates with the number of macrophages that accumulate [85]. Indeed, it has been demonstrated that macrophage infiltration is essential to promote FAP differentiation toward adipocytes and control the local fat deposition in the heart muscle [81]. On the contrary, no studies were addressed on the role of OPN in both FAP activation and cardiac macrophage infiltration, suggesting this issue as new starting point on the possible involvement of this pleiotropic protein on ectopic fat accumulation and scar deposition in the heart (Figure 1).

## 6. Renal–Fat Disease Ruled by OPN

The multiplicity of OPN functions also involve renal metabolism [26]. Indeed, OPN is physiologically expressed in kidneys and secreted into urine [26]. Normally, OPN is expressed in both fetal and mature kidneys in animals and humans [86]. OPN expression is present in most anatomical renal regions, although all of the OPN renal functions are not fully understood [86]. As in the heart, it seems that OPN has a “double-faced” role in the kidney as well, especially for its dual role in physiological and pathological signaling [87]. Only a few studies have highlighted OPN’s role in human tubulogenesis, discussing its involvement in the pro- and anti-role in stone and scar formation [1,87]. However, numerous studies have demonstrated that OPN mRNA and protein levels are augmented in many renal disorders, including interstitial fibrosis, stone formation, acute ischemic renal injury and many others [1,15]. Interestingly, all these studies report a direct correlation between the OPN protein level and fibrosis, macrophage infiltration, proteinuria and lowering creatine clearance, suggesting the potential use of OPN for different renal diseases [26]. Indeed, OPN is pathologically present with a dramatic increase in distal tubular cells, which are the principal cell type injured during tubulointerstitial nephritis, and in several models of acute ischemic renal damage [26]. From the scientific findings of an experimental model of a renal tumor, it is reported that OPN takes part also in kidney cancer inflammation [1], in which OPN resulted in being overexpressed, especially in tumor cells and host macrophages [1]. In addition, it is reported that in renal tumor biopsies, OPN^+^ macrophages are abundantly pronounced near the necrotic areas, suggesting the pivotal role of OPN in mediating pro-inflammatory signals in cancer [1,86]. Moreover, in decompensated and sclerotic vessels that underwent essential hypertension, OPN is more expressed and released by the tubules and its expression is related to α-smooth muscle actin upregulation in local renal fibroblasts, increasing the local progressive scar deposition [86]. In addition to the direct expression by renal cells during normal and pathological conditions, the main concomitant factor associated with the increase in OPN expression in the kidney is obesity [86].

## 7. Obesity Increase Renal OPN

Obesity is the predominant known risk factor for CKD and is directly related to incident renal function loss and mortality.

Intra-abdominal fat accumulation is related to a decline in fat mass over time and associated with CKD morbidity and mortality [86]. The body mass index (BMI), waist-to-hip ratio, waist-to-height ratio and conicity indexes are common and simple methods to measure VAT in dialysis patients and correlate them with the CKD prognosis [86,88,89]. Moreover, an increase in these indexes in CKD patients showed how obesity can adversely affect renal functions and promote renal damage, such as albuminuria [88]. Particularly, total body fat distribution is related to the albumin excretion rate (AER) [90], which increases more in CKD obese patients than in matched normal-weight patients [90]. The principal reason for renal decline is the low-grade inflammation response during obesity, which is amplified by macrophage infiltration both in the adipose and renal stroma [79]. Thus, in obese CKD patients, the kidneys are anatomically surrounded by a large quantity of fat cells and their drops are visible in histologically sections [86]. For this reason, the kidneys from obese people could be considered huge fat storing organs with a possible detrimental role in mediated low-grade renal inflammation [86]. Indeed, in renal fat, the pro-inflammatory M1 infiltrated macrophages promote the local release of IL-6, NFkB and adhesion molecules, including OPN [17,91]. Among the OPN^+^ macrophage functions in fat–renal diseases, the crucial role of the OPN^+^ macrophage infiltration in the conversion of calcium oxalate into renal stones is remarkable [87]. Indeed, in an in vitro model of cocultured renal tubular epithelial cells, 3T3-adipocytes and macrophages, OPN expression was triggered by a soluble factor released by the immune-infiltrate, which stimulated the other cocultured cells to overexpress OPN [87].

Moreover, it is also demonstrated that the 3T3-adipocytes can upregulate the OPN expression in cocultured macrophages, suggesting the intricated molecular interactions between the kidney and adipose tissue are ruled by the OPN^+^ immune infiltrate [87].

## 8. Conclusions

In summary, many aspects of the crossed pathophysiology of obesity-related disorders seem to be ruled by OPN signaling in the adipose tissue, heart and kidney (summarized in Figure 2). OPN can promote both adaptive and/or maladaptive signaling, controlling the macrophage activities in VAT and cardiac–renal stroma. The extended VAT seems to be the necessary and sufficient condition to invert the protective role of OPN in the targeted organs, amplify the detrimental properties of this intrinsically disordered protein. The pro-inflammatory switch of macrophages into the M1 state in VAT can also promote a cardiac and renal macrophage switch into the pro-inflammatory state. In the light of this, further studies should address the role of OPN in mediating inflammation in ectopic fat depots present at both the cardiac and renal level, to better understand the pivotal role of a fat mass increase in mediating inflammation thorough OPN.

## Figures and Tables

**Figure 1 ijms-21-05568-f001:**
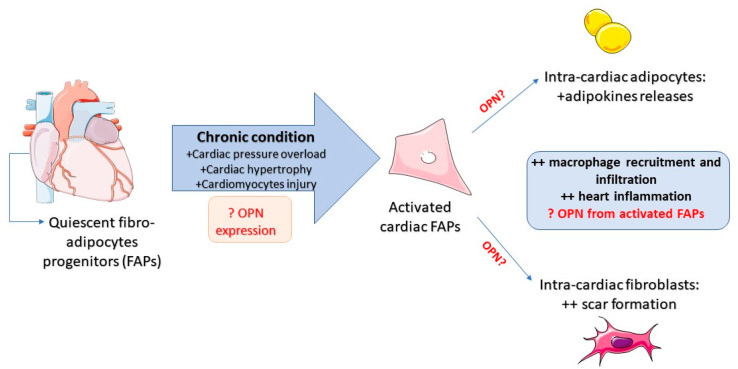
Cardiac fibro-adipocyte progenitor (FAP) activation in the heart: multipotent fibro-adipocyte progenitors (FAPs) are a stem reserve of the quiescent cells present in the heart, ready to be activated during heart injury. Normally, FAPs protect the myocardium by repairing lesions. Moreover, under chronic heart stresses, such as pressure overload, hypertrophy and cardiomyocyte injury, the quiescent FAPs become activated and differentiated in two non-cardiac lineages. Becoming intra-cardiac adipocytes, they release adipose mediators involved in heart inflammation, whereas becoming new intra-cardiac fibroblasts, they intervene in local collagen deposition and scar formation. Interestingly, both maturated lineages promote macrophage recruitment and activation, although no studies have underlined the possible involvement of osteopontin (OPN) in FAP activation and maturation.

**Figure 2 ijms-21-05568-f002:**
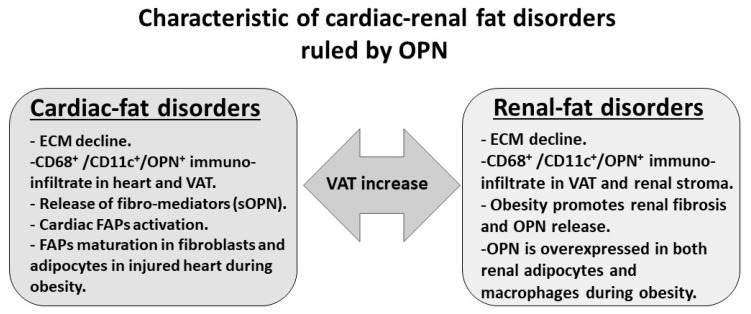
Summary diagram illustrating the role of OPN in mediating cardiac and renal disorders in the presence of obesity.

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
