# Peer review of "Osteopontin: The Molecular Bridge between Fat and Cardiac–Renal Disorders"

_ijms, 2020, doi:10.3390/ijms21155568_

Round 1
Reviewer 1 Report
This review proposes to integrate data concerning OPN adipose tissue to cardiac and kidney diseases. Many data are published about the role of osteopontin and association of its expression (mRNA or protein) with cardiac and kidney diseases. This work is interesting, however there is an overstatement of many reported results in the first paragraphs in order to try to convince reader and in many times it is only hypothesis from association studies. Moreover, the author must be careful in selected the right reference to argue its idea. Many references refer to other reviews that many times refer themselves to other review. Thus even it is a use to, the arguments are from an analysis and not the real data. The best is when it is possible, is to refer to original article.
The last part and conclusions are more convincing and more clearly make OPN a common key driver of CVD and KD.
More specifically:
Line 50 Ref 16 does not refer to cardiac or kidney-fat diseases but to total CVD in link to TD1
Line 119 OPN is not included in integrin family but some of their receptors do.
Line 120-121 I don’t think the reference 42 is relevant of the idea developed.
Line 154-157 “OPN mediate ECM responses” This sentence is an overstatement since in the previous line 153 it is indicated that OPN is both secreted and bound to ECM and there is no argument for a preferential function of bound OPN to attract macrophages. Moreover reference 51 is a review not referring specifically to OPN.
Line 168 this is your hypothesis, it is not argued.
Line 176 reference 57 and 58 refer to invitro study and are not relevant
Line 183 ref 58 and 61 not convincing for your argumentation (In vitro and review) 58 use RAW macrophage not ATM
Other minor concerns
In paragraph 5 appears the sub-paragraph 5.1 but there no 5.2. Remove numbering
Line 179 Hypertrophia in place of hypertropia
Author Response
Reply to Reviewer 1 (Rev#1)
This review proposes to integrate data concerning OPN adipose tissue to cardiac and kidney diseases. Many data are published about the role of osteopontin and association of its expression (mRNA or protein) with cardiac and kidney diseases. This work is interesting, however there is an overstatement of many reported results in the first paragraphs in order to try to convince reader and in many times it is only hypothesis from association studies. Moreover, the author must be careful in selected the right reference to argue its idea. Many references refer to other reviews that many times refer themselves to other review. Thus even it is a use to, the arguments are from an analysis and not the real data. The best is when it is possible, is to refer to original article.
The last part and conclusions are more convincing and more clearly make OPN a common key driver of CVD and KD.
- First, we would like to thank the Reviewer 1 for his/her useful suggestions which contributed to ameliorate our paper. Here we reply point-by-point his/her annotations, hoping that after text editing, our manuscript will be acceptable for publication in this prestigious Journal.
More specifically:
- Line 50 Ref 16 does not refer to cardiac or kidney-fat diseases but to total CVD in link to TD1.
As correctly highlighted by Rev#1 we changed the Ref 16 with these two new one:
- Francesca Schinzani et all numbered 24 in References section.
- Bostan G.O. et all numbered 21 in references section.
- Line 119 OPN is not included in integrin family but some of their receptors do.
As suggested by Rev#1 we modified the sentence as follows and all the modification are highlighted in red color on the text.
“As part of integrin receptors, OPN is the main mediators of ECM-cell interactions”.
- Line 120-121 I don’t think the reference 42 is relevant of the idea developed.
To better citing the sentence at line 120-121, we changed the reference 42 with this new one numbered 15 in the “References” section:
- Pollard, C. M.; Desimine, V. L.; Wertz, S. L.; Perez, A.; Parker, B. M.; Maning, J.; McCrink, K. A.; Shehadeh, L. A.; Lymperopoulos, A. Deletion of Osteopontin Enhances beta(2)-Adrenergic Receptor-Dependent Anti-Fibrotic Signaling in Cardiomyocytes, Int J Mol Sci 20(6) 10.3390/ijms20061396
- Line 154-157 “OPN mediate ECM responses” This sentence is an overstatement since in the previous line 153 it is indicated that OPN is both secreted and bound to ECM and there is no argument for a preferential function of bound OPN to attract macrophages. Moreover reference 51 is a review not referring specifically to OPN.
- To support our sentence we change our reference numbered 51 with the new 56, 57 and 58 references to better show the pivotal role of OPN on macrophages infiltration, using research article by Passmore M. et all (56), Li G. at all (57), and Trostel J. et all (58).
- Line 168 this is your hypothesis, it is not argued.
- As highlighted by Rev#1 we reported our personal hypothesis on OPN role in cardic-fat disorders linked to macrophage recruitment. For this reason, we reword the sentence as follows:
“In the light of this, it is plausible to consider OPN as a matricellular protein associated to cross-disorders linked to cardiac, renal fat’s decline and macrophages activation and recruitment in the site of tissue damage.
All the changes on the text are in red colour.
- Line 176 reference 57 and 58 refer to invitro study and are not relevant.
- As suggested, we deleted references 57-58 as considered it useless for the reader.
- Line 183 ref 58 and 61 not convincing for your argumentation (In vitro and review) 58 use RAW macrophage not ATM.
- Rev#1 highlighted that our references are not appropriate in sentence present in line 183. We reword the concept also chancing the citations with these one:
-Miriam Aouadi et all (2013) numbered 69 in “References” section.
- Xavier Prieur et all (2011) numbered 70 in “References” section.
The changes in text are in red.
Other minor concerns (Rev#1)
- In paragraph 5 appears the sub-paragraph 5.1 but there no 5.2. Remove numbering.
- As suggested by the Rev#1 we removed the sub-paragraph numbered 5.1.
- Line 179 Hypertrophia in place of hypertropia.
- As correctly noted, we changed “hypertropia” with the word “hypertrophy”.

Reviewer 2 Report
The manuscript by Vianello et al, entitled “Osteopontin: the molecular bridge between fat and cardiac-renal disorders” focuses on the role of osteopontin in obesity, cardiac-fat diseases, and renal-fat diseases. This concise review is a useful source of the previous research results and will help to understand the significance of osteopontin in those diseases. Nevertheless, the manuscript in its current form is not well organized and most references are not appropriate.
Major comments
- Authors mentioned iOPN in the Introduction section. However, the description of iOPN is unnecessary, as it is not directly relevant to the topic of this paper and all OPNs described in the manuscript are secreted one.
- Line 41: references 7, 8 are not appropriate for iOPN.
- Several sentences in section 2 are redundant, as they already appear in the introduction section. These sections of the paper require careful editing.
- Line 78: ‘These interactions are mediated by the two different terminal zones generated by thrombin or metalloproteinases cleavage, aimed to bind different ligands.’ This sentence is misleading because it suggests that only OPN fragments generated by proteinase cleavage can bind ligands. Indeed, full-length OPN also can interact with receptors.
- Line 39: The term, ‘matricellular protein’ will be unfamiliar to some readers and requires a brief description. The first paragraph in the section 3 should be included in the introduction section.
- Line 119: ‘As part of integrin family, OPN is the main mediators of ECM-cell interactions. The binding between ……in targeted tissues ’ I am not convinced these sentences are correct. Please show more appropriate references.
- Line 135-139: Please refer to original papers instead of review 44.
- Line 139: ‘OPN is also considered…….or resorption [43]’ I cannot find the description in ref 43. Please refer to appropriate references.
- Line 149: There is no description in ref 48.
- Line 217: Is ref 34 correct to the sentence?
- Line 331: ‘this structural protein’ is not correct because OPN is a matricellular protein and also a complete intrinsic disorder protein.
Minor comments
- Line 63: ‘uroprotein of T lymphocyte activator-1 (Eta-1)’ should be ‘early T lymphocyte activator-1 (Eta-1)’.
- Line 143: What does ‘that’ mean?
- There are several typos in the manuscript. For example on page 4, line 152 ‘EMC’ should be ‘ECM’. Page 7 line 283, ‘tubologenesis’ should be ‘tubulogenesis’. Please proofread the manuscript carefully.
Author Response
Reply Letter for our Manuscript ID: ijms-835281 by Vianello et all
titled “Osteopontin: the molecular bridge between fat and cardiac-renal disorders”
Reply to Reviewer 2
The manuscript by Vianello et al, entitled “Osteopontin: the molecular bridge between fat and cardiac-renal disorders” focuses on the role of osteopontin in obesity, cardiac-fat diseases, and renal-fat diseases. This concise review is a useful source of the previous research results and will help to understand the significance of osteopontin in those diseases. Nevertheless, the manuscript in its current form is not well organized and most references are not appropriate.
We would like to thank you the Reviewer 2 (Rev#2) for his/her suggestions needed to improve the text message of our Review.
Major comments
- Authors mentioned iOPN in the Introduction section. However, the description of iOPN is unnecessary, as it is not directly relevant to the topic of this paper and all OPNs described in the manuscript are secreted one.
We agree with the Rev#2 that iOPN is not discussed in the rest of full body text, but we think that could be a possible suggestion for the reader for new study on OPN signaling, because really few it is known about OPN regulation and cardiac/renal fat disorders. For this reason, we would like to maintain this concept in Introduction. If the Reviewer 2 disagree with us about this, we can remove it.
- Line 41: references 7, 8 are not appropriate for iOPN.
As indicated by Rev#2 we add the new reference numbered 6 in “References” section more appropriate to discuss iOPN role in the text.
- Several sentences in section 2 are redundant, as they already appear in the introduction section. These sections of the paper require careful editing.
The redundant concepts presented both in Introduction and in section 2 are maintained only in section 2 because more appropriate with the concept of this paragraph.
For this reason, we remove this redundancy from Introduction.
We deleted from line 66 to 71 as follows:
“Among them, sOPN can interact, for example, with macrophage receptors, like CD44, enhancing the release of interleukin (IL)-12 and promoting the development of T helper cells[5, 16, 17]. Interestingly, OPN can also mediate ectopic calcification signaling in non-bone cells, like vascular smooth muscle cells, regulating their intracellular factors, including bone morphogenic proteins (BMPs) which can induce a shift of lineage to osteoblast-like cells, promoting calcium deposition[18-20].”
Since Rev#2 suggested to move 1st paragraph of section 3 in Introduction section, we deleted in this section the redundant concept common with section 2 to ameliorate the final version of both paragraphs.
We hope that these changes are also suitable for Rev#2.
All the changes are in red colour in the text.
- Line 78: ‘These interactions are mediated by the two different terminal zones generated by thrombin or metalloproteinases cleavage, aimed to bind different ligands.’
This sentence is misleading because it suggests that only OPN fragments generated by proteinase cleavage can bind ligands. Indeed, full-length OPN also can interact with receptors.
As suggested, we reword the sentence at line 78 to explicate better the correct concept. We added the adverb “principally” to lose not the full-length OPN interactions.
The change is in red colour in the text.
- Line 39: The term, ‘matricellular protein’ will be unfamiliar to some readers and requires a brief description. The first paragraph in the section 3 should be included in the introduction section.
As suggested by Rev#2 we moved the first paragraph of section 3 at Introduction, to line 41 to 64, to introduce the matricellular protein in the text as unfamiliar concept for some readers.
All the changes are in red color.
- Line 119: ‘As part of integrin family, OPN is the main mediators of ECM-cell interactions. The binding between ……in targeted tissues’ I am not convinced these sentences are correct. Please show more appropriate references.
As highlighted by Rev#2 we reword the sentence at line 119 to make it more appropriated and correctly cited.
- Line 135-139: Please refer to original papers instead of review 44.
As suggested by Rew#2 we modified the citation with these new original articles numbered as follows in “References” section:
-Lancha et all PlosOne 2014 as new 46 citation.
-Sawaki D et all Circulation 2018 as new 29 citation.
-Schunch K et all, Obesity 2016 as new 47 citation.
All the changes are in red colour.
- Line 139: ‘OPN is also considered…….or resorption [43]’ I cannot find the description in ref 43. Please refer to appropriate references.
As suggested, we reported new appropriate reference numbered 48 in “Reference” section.
- Line 149: There is no description in ref 48.
As correctly highlighted, we changed the ref 48 with the new one 53 in References sections.
- Line 217: Is ref 34 correct to the sentence?
Yes, it is.
- Line 331: ‘this structural protein’ is not correct because OPN is a matricellular protein and also a complete intrinsic disorder protein.
As suggested by the Rev#2 we deleted “this structural protein” in the conclusion sentence and we add his/her suggestion as he/she can see in the conclusion section.
Minor comments Rev#2
- Line 63: ‘uroprotein of T lymphocyte activator-1 (Eta-1)’ should be ‘early T lymphocyte activator-1 (Eta-1)’.
As correctly highlighted by Re#2 we add ‘early’ both in “introduction” and in “abbreviation” sections.
- Line 143: What does ‘that’ mean?
The sentence would indicate that OPN could promote the turnover of ECM in fat but also which of cardiac and renal tissues.
- There are several typos in the manuscript. For example on page 4, line 152 ‘EMC’ should be ‘ECM’. Page 7 line 283, ‘tubologenesis’ should be ‘tubulogenesis’. Please proofread the manuscript carefully.
We carefully control our manuscript, deleting typos like which one in line 283.

Round 2
Reviewer 1 Report
The author have answered to all my concerns but one is still to improve
at line 57 I propose to replace “As part of integrin receptors family, OPN is the main mediators of ECM-cell interactions[15]” by “As part of integrin and CD44 ligands, OPN is one of the main mediators of ECM-cell interactions[15]”..
And replace ref 15, I think for this general idea the following ref that really detailed many of the signaling below theses receptors would be better. However other reference less specific than the current 15, would be great.
David T. Denhardt, … , Dubravko Pavlin, Jeffrey S. Berman. Osteopontin as a means to cope with environmental insults: regulation of inflammation, tissue remodeling, and cell survival. J Clin Invest. 2001;107(9):1055-1061. https://doi.org/10.1172/JCI12980.
A least take care of reference numbering because erasing of some references lead to removing of the number 23, 65, 68
Author Response
Response to Reviewer 1 Comments
Manuscript ID ijms-835281
The author have answered to all my concerns but one is still to improve.
Point 1: at line 57
I propose to replace “As part of integrin receptors family, OPN is the main mediators of ECM-cell interactions[15]” by “As part of integrin and CD44 ligands, OPN is one of the main mediators of ECM-cell interactions[15]”..
And replace ref 15, I think for this general idea the following ref that really detailed many of the signaling below theses receptors would be better.
However other reference less specific than the current 15, would be great.
David T. Denhardt, … , Dubravko Pavlin, Jeffrey S. Berman. Osteopontin as a means to cope with environmental insults: regulation of inflammation, tissue remodeling, and cell survival. J Clin Invest. 2001;107(9):1055-1061. https://doi.org/10.1172/JCI12980.
Response 1: As suggested by the Reviewer 1 we modified the sentence as recommended and we changed the previous ref 15 with the new ref 11. All the changes are present in red colour at line 47.
Point 2: A least take care of reference numbering because erasing of some references lead to removing of the number 23, 65, 68.
Response 2: As highlighted by the Reviewer 1, we controlled all the References after manuscript drafting during second step of revision phase.
Reviewer 2 Report
- Line 41: Matricellular proteins are components of ECM. Therefore, “matricellular components” are curious expression. The explanation in Lin41-64 is difficult to understand because the explanations are mixtures of matricellular protein and ECM and some explanations are inappropriate. The authors should read more general review about OPN. I also suggest that the authors try to shorten it. I recommend that the authors read the following reviews and should rewrite the sentences.
Morris, 2014 Matrix Biology, p183 and Murphy-Ullrich 2014 Matrix Biology, p1.
- Line 41: ……..different cell activities. What kinds of cell activities? Please be more specific about them.
- In introduction section: I think the description of iOPN is unnecessary because there is no description about the relationship between iOPN and cardiac/renal fat disorders and readers are confusing.
- Line 102. It is still misleading. “generated by thrombin or metalloproteinases cleavage” should be deleted from the sentence.
- Line 57: the statement “OPN is the main mediators of ECM-cell interactions” is too strong and should be rewritten as, “OPN is one of the main mediators of ECM-cell interactions”. I cannot understand “ as part of integrin receptors family”. What does it mean? Ref 15 is not appropriate for this sentence.
- Line 335: “intrinsic disorder protein” should be “intrinsically disordered protein”.
- Line 139: ref 29 is not appropriate.
- Line 44: Is ref44 correct? Ref48?
9. Line 145 (previous 143): In response to Minor comment 2, the authors stated that “the sentence would indicate that OPN could promote the turnover of ECM in fat but also which of cardiac and renal tissues.” This expression is easy to understand compared to original sentence. Therefore, the sentence should be ex
Author Response
Response to Reviewer 2 Comments
Manuscript ID ijms-835281
Point 1: Line 41: Matricellular proteins are components of ECM. Therefore, “matricellular components” are curious expression.
The explanation in Lin41-64 is difficult to understand because the explanations are mixtures of matricellular protein and ECM and some explanations are inappropriate.
The authors should read more general review about OPN.
I also suggest that the authors try to shorten it.
I recommend that the authors read the following reviews and should rewrite the sentences.
Morris, 2014 Matrix Biology, p183
Murphy-Ullrich 2014 Matrix Biology, p1.
Response 1: We thank you the Reviewer 2 for his/her detailed annotations. We agree with his/her suggestions and after reading the Reviews recommended, we rewritten the paragraph focused on the introduction of matricellular proteins, shortening it as indicated.
All the changes are from lines 37 to 51 in red color.
Point 2: Line 41: ……..different cell activities.
What kinds of cell activities? Please be more specific about them.
Response 2: As requested, we add the major cell activities in which OPN is involved.
We added this sentence “including cell proliferation and inflammatory responses”in line 41 in red colour.
Point 3: In introduction section: I think the description of iOPN is unnecessary because there is no description about the relationship between iOPN and cardiac/renal fat disorders and readers are confusing.
Response 3: As suggested by the Reviewer 2, we deleted the concept of iOPN both in “Introduction” and “Abbreviations” sections and in Graphical Abstract.
The modification is in red colour in the text.
Point 4: Line 102. It is still misleading.
“generated by thrombin or metalloproteinases cleavage” should be deleted from the sentence.
Response 4: as suggested by the Reviewer 2, we deleted this sentence in line 85 (previous 102).
Point 5: Line 57: the statement “OPN is the main mediators of ECM-cell interactions” is too strong and should be rewritten as,
“OPN is one of the main mediators of ECM-cell interactions”. I cannot understand “as part of integrin receptors family”.
What does it mean? Ref 15 is not appropriate for this sentence.
Response 5: This point is also debated by other Reviewer who suggested to reword the sentence as follows: “As part of integrin and CD44 ligands, OPN is one of the main mediators of ECM-cell interactions.”, recommended this citation “David T. Denhardt, … , Dubravko Pavlin, Jeffrey S. Berman. Osteopontin as a means to cope with environmental insults: regulation of inflammation, tissue remodeling, and cell survival. J Clin Invest. 2001;107(9):1055-1061. https://doi.org/10.1172/JCI12980”, which should replace the previous ref. 15. as new ref. 11
Point 6: Line 335: “intrinsic disorder protein” should be “intrinsically disordered protein”.
Response 6: As suggested we changed the conclusion text as recommended at new lines 315-316 (previous 335).
Point 7: Line 139: ref 29 is not appropriate.
Response 7: As suggested by the Reviewer 2 we removed only the ref 29 in previous line 139 new 122.
Point 8: Line 44: Is ref 44 correct? Ref48?
Response 8: Because Reviewer 2 suggested to us to short the paragraph focused on matricellular protein and OPN, both these references are removed.
Point 9: Line 145 (previous 143): In response to Minor comment 2, the authors stated that “the sentence would indicate that OPN could promote the turnover of ECM in fat but also which of cardiac and renal tissues.”
This expression is easy to understand compared to original sentence.
Therefore, the sentence should be ex.
Response 9: We have not understood what the Reviewer 2 means in his/her suggestion “Therefore, the sentence should be ex”.
If the Reviewer 2 suggests to explain better the concept, we think that in the subsequent sentences we developed better this “introductory” concept.